# Charting the Ethical Frontier in Newborn Screening Research: Insights from the NBSTRN ELSI Researcher Needs Survey

**DOI:** 10.3390/ijns10030064

**Published:** 2024-09-19

**Authors:** Yekaterina Unnikumaran, Mei Lietsch, Amy Brower

**Affiliations:** 1American College of Medical Genetics and Genomics (ACMG), Bethesda, MD 20814, USA; yunnikumaran@gmail.com (Y.U.); mei-lietsch@uiowa.edu (M.L.); 2Carver College of Medicine, University of Iowa, Iowa City, IA 52242, USA; 3Genetic Medicine, Munore-Meyer Institute, University of Nebraska Medical Center, Omaha, NE 68106, USA; 4Department of Health Professions., Creighton University School of Medicine, Omaha, NE 68178, USA

**Keywords:** newborn screening, ethical, legal, social issues, ELSI, research, rare disease, NBSTRN, privacy, informed consent, public health

## Abstract

From 2008 to 2024, the Newborn Screening Translational Research Network (NBSTRN), part of the National Institute of Child Health and Human Development (NICHD) Hunter Kelly Newborn Screening Program, served as a robust infrastructure to facilitate groundbreaking research in newborn screening (NBS), public health, rare disease, and genomics. Over its sixteen years, NBSTRN developed into a significant international network, supporting innovative research on novel technologies to screen, diagnose, treat, manage, and understand the natural history of more than 280 rare diseases. The NBSTRN tools and resources were used by a variety of stakeholders including researchers, clinicians, state NBS programs, parents, families, and policy makers. Resources and expertise for the newborn screening community in ethical, legal, and social issues (ELSI) has been an important area of focus for the NBSTRN and this includes efforts across the NBS system from pilot studies of candidate conditions to public health implementation of screening for new conditions, and the longitudinal follow-up of NBS-identified individuals to inform health outcomes and disease understanding. In 2023, the NBSTRN conducted a survey to explore ELSI issues in NBS research, specifically those encountered by the NBS community. Since NBS research involves collaboration among researchers, state NBS programs, clinicians, and families, the survey was broadly designed and disseminated to engage all key stakeholders. With responses from 88 members of the NBS community, including researchers and state NBS programs, the survey found that individuals rely most on institutional and collegial resources when they encounter ELSI questions. Most survey responses ranked privacy as extremely or very important in NBS research and identified the need for policies that address informed consent in NBS research. The survey results highlight the need for improved collaborative resources and educational programs focused on ELSI for the NBS community. The survey results inform future efforts in ELSI and NBS research in the United States (U.S.) and the rest of the world, including the development of policies and expanded ELSI initiatives and tools that address the needs of all NBS stakeholders.

## 1. Introduction

The landscape of medical research is underscored by ever-present ethical, legal, and social issues (ELSI). As a fundamental aspect of all scientific inquiries, ELSI considerations play an indispensable role in guiding the ethical conduct, legal framework, and social implications of research endeavors. Although consideration of ELSI is firmly established across the medical research spectrum [1], their integration into the realm of newborn screening (NBS) research, which inherently requires collaboration between academic, clinical, public health, patient, and family communities, remains an area in which more cohesive and comprehensive implementation is needed [2,3,4].

For 16 years, the Newborn Screening Translational Research Network (NBSTRN) was a robust infrastructure dedicated to facilitating and expanding groundbreaking NBS research [5,6]. Founded in 2008 as a key component of the Hunter Kelly Newborn Screening Program at the Eunice Kennedy Shriver National Institutes of Child Health and Human Development (NICHD) through a contract with the American College of Medical Genetics and Genomics (ACMG), NBSTRN grew into an international network involved in supporting cutting-edge research, population-based pilots, and longitudinal studies to discover novel technologies to screen, diagnose, treat, and manage rare disease across the lifespan [7,8,9,10,11,12,13]. NBSTRN tools and resources have been used by a variety of stakeholders, including researchers, clinicians, state NBS programs, parents, families, and policy makers working across the NBS system of basic, translational, and clinical research, prenatal education, neonatal screening, clinical diagnosis, treatment, and lifelong care [14,15,16,17,18,19,20,21].

Resources and expertise for the NBS community in ethical, legal, and social issues (ELSI) has been an important area of focus for the NBSTRN, and this includes efforts across the NBS system from pilot studies of candidate conditions to public health implementation of screening for new conditions, and the longitudinal follow-up of NBS-identified individuals to inform health outcomes and disease understanding [4,22,23]. In 2023, to better understand ELSI issues in NBS research encountered by the NBS community, the NBSTRN conducted a survey of the NBS community. Because NBS research involves collaboration between researchers, state NBS programs, clinicians, and families, the survey was designed for all stakeholders and widely disseminated to reach all these groups. We describe the survey design, execution, results, and key takeaways to inform future efforts in ELSI and NBS research in the United States (U.S.) and the rest of the world.

### 1.1. ELSI across the NBS System

In the U.S., NBS is led by the state public health departments and is recognized as one of the most triumphant public health initiatives. Since the inception of the first successful screen for phenylketonuria (PKU) in the 1960s, state NBS programs have expanded their list of conditions to include screenings for up to 81 conditions [24], with the goal of facilitating early intervention for affected newborns. The decision-making power regarding condition inclusion within NBS programs lies with state NBS programs, and the makeup of screening panels are informed by the Recommended Uniform Screening Panel (RUSP). The RUSP is a list of disorders that the Secretary of the Department of Health and Human Services (HHS) recommends for states to screen. A federal advisory committee coordinates a process of nomination, evidence review, and recommendation which evaluates information provided by nominators and assesses the net benefit of screening all newborns, the certainty of the evidence regarding the net benefit, the feasibility of implementing a comprehensive program of screening for the condition, and the readiness of public health programs to implement such a program of expanded screening, including an assessment of costs to the newborn screening programs to expand screening for a condition under review [25,26,27].

The past two decades have witnessed a rapid expansion in NBS capabilities fueled by technological advancements in tools to screen and diagnose, including sequencing, and novel approaches to treatment, including precision medicine [15,19,28,29,30]. This technological expansion and complexity of NBS research underscore the urgent requirement to address ELSI considerations across each component of NBS from parental education, sample collection and screening, communication of results, confirmatory testing, treatment, and long-term follow-up [31,32]. This is particularly critical given the multi-dimensional scope of NBS research, which includes pilot studies to evaluate candidate conditions for the RUSP nomination, review, and public health implementation [33,34,35,36,37,38]. The RUSP process reveals the intricate relationship between ELSI elements and NBS, as well as the interplay between researchers, state NBS programs, clinicians, and families. As a whole, this prioritizes ELSI as a crucial component of the NBS system. This includes the differences in disease expression, symptoms, severity, and outcomes resulting from population-based screening; the preparedness of state programs for screening expansion; the increasing complexity of care coordination for conditions with varying clinical onsets and treatment urgency; the integration of trauma-informed care to support families impacted by the screening process, and equitable access to timely diagnosis, treatment, and life-long care and management regardless of the location of birth [39,40,41,42].

### 1.2. NBSTRN Efforts to Facilitate ELSI in NBS

In the U.S., the RUSP is an established a system of nomination, review, and endorsement to establish a list of conditions recommended for NBS [24]. The evidence review includes an assessment of the potential benefits and harms of NBS informed by NBS research that includes pilot and natural history studies led by academia, public health, and advocacy groups. NBSTRN served as a coordinating center for several pilot and natural history studies and found that most pilot studies involved collaboration between researchers, clinicians, and state NBS programs. We observed that the approaches to informed consent, institutional board review (IRB), and consumer input varied across pilot sites [9,10]. In 2019, the NBSTRN Bioethics and Legal Workgroup (WG) conducted a series of professional and public discussions to begin to address the observed differences in pilot studies with the goal of identifying important ELSI challenges and facilitating the integration of ELSI into NBS pilot studies [4]. This effort identified nine key ELSI questions that could be integrated into pilot studies to ultimately help the NBS program to better understand the potential impact of screening for a new condition on newborns and families. To facilitate the continued exploration of ELSI in NBS, we designed a survey in 2023 to gather input on ELSI needs in NBS research for all stakeholders in the NBS community.

## 2. Materials and Methods

We developed an online survey to assess the ELSI challenges in NBS research and identify the resources that are used or of interest to the NBS community, including researchers, state NBS programs, clinicians, advocacy groups, and individuals living with and parents of an individual with a disease identified through NBS. The survey was designed using a series of online meetings and communications with the NBSTRN Bioethics and Legal Workgroup and consisted of one open-ended question and thirteen multiple choice questions written in English. Identifying information was not included in the survey. The survey was developed, validated, and disseminated using Research Electronic Data Capture (REDCap) ([43], Appendix). An electronic link to the survey was disseminated to a convenience sample consisting of the NBSTRN network that included individuals who had registered for the NBSTRN website, participated in events, and received the NBSTRN newsletter. The survey was deployed during the NBSTRN Network Meeting on 18 May 2023, and this was followed by a comprehensive outreach campaign that included social media engagement, directed emails to the NBSTRN network members, and listserv announcements. To ensure a wide reach within the NBSTRN community, broader dissemination efforts continued, and the survey remained open until 25 August 2023. The Accumulated data were exported from REDCap^®^ (https://projectredcap.org/software/, accessed on 15 August 2024), and the percentage of participants who selected each response was computed for the thirteen binary and/or categorical response options. Three staff members independently reviewed the text responses for the open-ended question and identified themes.

The privacy and confidentiality of the survey respondents were prioritized throughout the study. Personal identifiers were removed from the dataset to protect the anonymity of the participants. Data were stored securely and only accessible to authorized personnel directly involved in the study.

## 3. Results

Eighty-eight (*N* = 88) individuals accessed the 13-question survey, and a data quality check found that 98% (86/88) answered all questions. Incomplete surveys were included in the analysis for the questions that were completely answered. Figure 1 presents the results of twelve multi-choice questions (Q1–11, Q13) using GraphPad Prism 6 [44]. The themes identified from the open-ended question (Q12) are listed in Table 1.

Of those who accessed the survey, 36% (32/88) identified their role in NBS research as researcher, 27% as state program (24/88%), 15% (13/88) as clinical care, and 9% (8/88) as advocacy (Figure 1, Q2). A total of 81% (71/88) has been involved in NBS research for three or more years, and 86% (75/88) identified privacy as an important ethical issue in NBS research (Figure 1, Q4, Q12). When encountering legal or ethical issues related to NBS research, 35% (31/88) typically seek information or guidance from institutional lawyers or ethics committees within their organization, 24% (21/88) utilized colleagues, 18% (16/88) other resources including the literature, 11% (10/88) nonprofit organizations, and 9% (8/88) utilized state lawyers with expertise in healthcare law (Figure 1, Q5). A total of 63% (55/88) were aware of the NBSTRN ELSI resources, and 80% (60/88) were somewhat or very interested in a research ethics consultation service that helps to answer ethical issues around the ethical issues surrounding NBS research (Figure 1, Q6, Q15).

To examine differences between the individual stakeholder groups related to ELSI information or guidance, we compared the responses to Q5, “When encountering legal or ethical issues related to newborn screening research, where do you typically seek information or guidance”, by self-identified stakeholder groups: 55% (16/29) of self-identified researchers sought information or guidance from colleagues conducting ethical research. A total of 58% (14/24) of self-identified state NBS program members sought information or guidance from institutional lawyers or ethics committees within their organization and 25% (6/24) consulted with state lawyers specialized in healthcare law. A total of 46% (6/13) self-identified clinicians sought guidance from institutional lawyers.

Forty-seven percent (41/88) of participants answered the open-ended question (Question 12; Appendix A): “What are the ethical, legal, or social issues that you face in your own work? Explain.” Seven common themes were identified by three staff members who reviewed the text responses. Table 1 details the seven themes and the frequency with which they were mentioned by the participants. Informed consent was a theme identified by 34% (14/41) participants.

## 4. Discussion and Future Efforts

NBS research is a crucial area that aims to presymptomatically identify and treat rare, often life-threatening conditions early in a child’s life. This research is essential because early intervention can significantly improve health outcomes and quality of life through the discovery of novel technologies to screen, diagnose treat, and manage disease. Given the complexity and rarity of these diseases, coupled with population-based screening that is coordinated by state NBS programs, collaboration among various stakeholders to plan, carry out, disseminate, and implement research findings is critical. Including the consideration of ELSI in the NBS research is also important and our survey of a convenience sample of the NBS community found overwhelming interest in training or educational activities related to ethical issues in NBS research and identified a need for improved collaborative frameworks, tools and educational programs focused on ELSI for the NBS community. Sixty-three percent (56/88) of respondents self-identified as either researchers or state NBS programs. This reinforces the experience of many that research in NBS involves both collaboration between researchers and state NBS programs. Therefore, to address ELSI in NBS research, we should consider the concerns and needs that these groups have in common as well as those that are unique to researchers based in academia or foundations, public health team members working in state departments of health, and clinical care providers in hospitals and community based centers.

Across all stakeholder groups, the participants identified privacy and informed consent as key issues to address in NBS research. Almost all (95%, 83/88) respondents were somewhat or very interested in a research ethics consultation service focused on the ethical issues surrounding NBS research. The preference for educational formats that facilitate direct engagement and real-time discussion reveals a potential gap in the current educational offerings that are focused on static content. In summary, the survey results inform future efforts in ELSI and NBS research in the U.S. and have relevance to the rest of the world, including the development of policies and expanded ELSI frameworks and tools that address the needs of all NBS stakeholders.

These future efforts to support ELSI in NBS research will be important as the number of diseases, treatments, and technologies increase and the complexities surrounding ELSI issues continue to grow. In addition, groups around the world are researching ways to integrate genomics to screen, diagnose, and design interventions in newborns and children, and this heightens the need for robust, coordinated, and widely disseminated ELSI efforts and policies [45]. Informed by the sixteen years of discussions, collaborations, and workgroups of the NBSTRN, Table 2 lists nine suggested key areas for future efforts and provides examples to illustrate how targeted projects can address ELSI considerations in NBS research by leveraging patient-centric resources, participatory research models, public education and engagement, and collaborative education programs. The NBSTRN of diverse stakeholders across the NBS community from public health partners to clinicians, patients, parents, families, researchers, and advocacy groups continually highlighted the importance of a focus on privacy, informed consent, interdisciplinary collaboration, social acceptance, and ongoing education. This broad approach to ELSI in NBS research coupled with innovative initiatives that consider local, regional, and county-wide policies and practices will help to ensure that ELSI aspects are comprehensively integrated into NBS research around the world. In the U.S., the continued collaboration between the National Institutes of Health (NIH), the Centers for Disease Control and Prevention (CDC), and the Health Resources and Services Administration (HRSA) will be vital in supporting these efforts to advance ELSI in NBS research and ensure the successful implementation of evolving best practices. Each of these agencies brings unique strengths and expertise to the table, creating a comprehensive approach that addresses scientific discovery, translational research, public health implementation and surveillance, and clinical and community-based care across the lifespan delivery aspects [46]. In conclusion, the NBSTRN ELSI survey results demonstrate that the NBS community recognizes the important of ELSI in NBS research infrastructure and calls upon all stakeholders and federal partners to invest in a future where rare disease research is prioritized and supported across research, clinical care, and state NBS programs.

## Figures and Tables

**Figure 1 IJNS-10-00064-f001:**
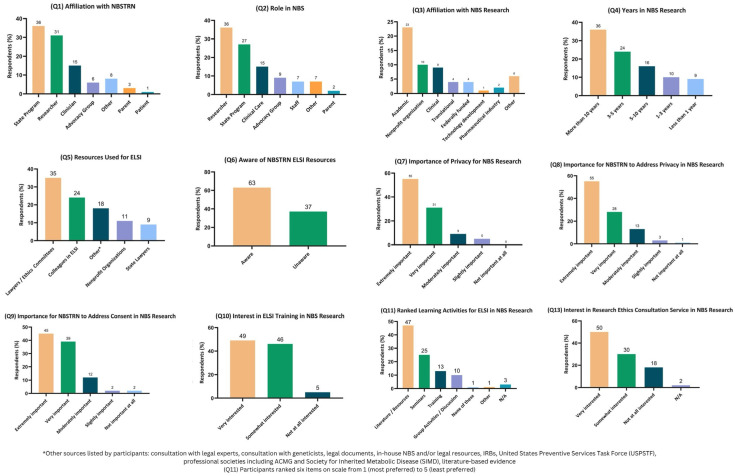
Results from multiple choice questions Q1 to Q11 and Q13 (the results are reported in descending order, from left to right). Not Applicable (N/A).

**Table 1 IJNS-10-00064-t001:** Areas for ELSI Considerations identified in the open-ended question.

Theme	Number of Mentions (*n* = 41)	Percentage of All Responses
Informed consent	14	34%
Usage and standardization of data	9	22%
Usage and storage of dried blood spots (DBS)	6	15%
Policy	6	15%
The NBS process	6	15%
Privacy	4	10%
Population diversity of data and samples	4	10%

**Table 2 IJNS-10-00064-t002:** Nine suggested areas for future efforts to advance ELSI in NBS research for the worldwide NBS community.

Key Area	Description	Example(s)
ELSIs in NBS Research	Funded efforts to develop, validate, pilot, and implement accessible, patient-centric resources that address the burgeoning ELSI challenges identified by the NBS research community. Emphasis would be placed on privacy and informed consent tools, catering to the nuanced requirements of rare disease research.	Establish a global consortium dedicated to addressing ELSIs specific to genomic newborn screening. The consortium would develop guidelines, share best practices, and support collaborative research projects. This could be used to develop an online platform called the “Patient-Centric ELSI Resource Hub” to create, validate, pilot, and implement resources addressing ELSI challenges in NBS. Key features are privacy and informed consent tools, personalized education materials for parents, interactive FAQs, and case studies tailored to rare diseases. Users include researchers, clinicians, parents, and patient advocacy groups. This hub would ensure that parents are well-informed about privacy and consent issues, and it would help researchers address ELSI challenges effectively, promoting ethical and transparent research practices.
Innovative NBS Research Approaches	Design NBS research using a participatory research model that involves parents and communities in the design and implementation of research studies ensuring that their perspectives and needs are considered, an incorporating trauma-informed care to address emotional impacts. Forming interdisciplinary research teams that include ethicists, legal scholars, sociologists, and community representatives to address ELSI issues comprehensively. Developing tools and methodologies to assess the impact of ELSI considerations in NBS programs, ensuring continuous improvement and responsiveness to emerging challenges.	Create councils consisting of patients and family members affected by conditions identified through NBS and NBS research. These councils would provide insights into real-world impacts of screening programs and help to guide ELSI policies and practices. Establish a consortium called the “Participatory NBS Research Consortium” that uses a participatory research model involving parents and communities in the design and implementation of NBS research. Activities focus on engaging parents and community representatives in research design, forming interdisciplinary teams with ethicists, legal scholars, sociologists, and community representatives and developing assessment tools for ELSI impact. Researchers, community representatives, ethicists, legal scholars, and sociologists would be members. This consortium would ensure that research studies are designed with input from those directly affected, leading to more ethically sound and socially acceptable research outcomes.
Enhancing Social Acceptance and Engagement	Developing comprehensive education and outreach programs to inform the public about the benefits, risks, and ethical considerations of newborn screening, fostering a greater acceptance and informed participation. Providing cultural competence training for healthcare providers to ensure sensitive and respectful communication with families from diverse backgrounds. Creating platforms for public engagement where stakeholders, including parents, patient advocacy groups, and the public, can discuss and influence newborn screening policies and practices.	Launch nationwide and worldwide campaigns to educate the public about NBS research, focusing on ELSI implications. Use multimedia platforms, community events, and partnerships with advocacy groups to research diverse audiences and create the “Newborn Screening Education and Outreach Initiative”. Activities include creating educational materials and campaigns, cultural competence training for healthcare providers, and establishing public engagement platforms for discussion and policy influence. Collaborators include healthcare providers, educators, patient advocacy groups, and public health officials. This initiative would foster a greater public understanding and acceptance of NBS, ensure respectful and sensitive communication with diverse families, and provide a forum for public input on NBS policies.
Collaborative Education Programs	With an expressed preference for interactive and structured learning found in our survey, supporting the establishment of seminars and workshops. These would serve to disseminate current ELSI practices and foster a collaborative learning environment, integrating patient advocacy groups in the educational design to ensure patient-relevant outcomes.	Establish dedicated research centers focused on studying the ELSI aspects of NBS research. These centers would conduct interdisciplinary research, provide training for healthcare professionals, and offer policy recommendations. Develop a series of seminars and workshops called the “ELSI in NBS Seminar Series” to disseminate current ELSI practices and foster a collaborative learning environment. The format consists of interactive sessions with case studies, panel discussions, and breakout groups, co-designed with patient advocacy groups to ensure relevance and impact. Participants include researchers, clinicians, patient advocates, legal experts, and ethicists. This seminar series would promote ongoing education on ELSI issues, facilitate collaboration across disciplines, and ensure that patient perspectives are integrated into educational efforts.
Digital Engagement Platforms	Investments could be allocated towards developing digital platforms for ELSI education and discussion. These platforms would encourage cross-disciplinary collaboration and facilitate real-time dialogue among researchers, clinicians, and patient groups.	A dedicated online platform called the “Newborn Screening ELSI Hub” could be developed to provide a space for education and discussion around the ethical, legal, and social implications of newborn screening. The platform would include forums for real-time dialogue, webinars hosted by experts in the field, interactive case studies, and a repository of educational materials, such as articles, videos, and guidelines. Researchers, clinicians, ethicists, patient advocacy groups, and parents could be the targeted users. This platform would facilitate cross-disciplinary collaboration, ensuring that diverse perspectives are considered in ELSI discussions and decision-making processes.
Ethics Consultation Services	Recognizing the need for ongoing support in navigating ethical complexities, programs such as the Rare Diseases Clinical Research Network (RDCRN) could establish a consultative service within its consortia. This service would be tasked with providing expert advice on ethical considerations in clinical trial design and implementation, especially for rare diseases where the ethical landscape can be particularly complex.	The Rare Diseases Clinical Research Network (RDCRN) could establish an Ethical Consultation Unit within its consortia to provide expert advice on ethical issues in clinical trials, particularly those involving rare diseases. The unit would offer consultations on trial design, informed consent processes, data management, and the handling of incidental findings. It would also provide training sessions for researchers and support for navigating ethical review boards. The unit would comprise bioethicists, legal experts, patient advocates, and experienced clinicians. This service would ensure that ethical considerations are thoroughly addressed in the planning and execution of clinical trials, enhancing the integrity and acceptability of research projects.
Legal and Regulatory Navigation Tools	Considering the diverse regulatory environments encountered across research sites, funding could be targeted to the development of tools that aid rare disease researchers in understanding and complying with local and international regulations, thereby ensuring the ethical conduct of rare disease research.	The RDRN could be a comprehensive digital tool designed to help researchers to understand and comply with varying legal and regulatory requirements across different regions. Interactive maps showing regulatory landscapes, step-by-step guides for compliance, templates for necessary documentation, and a database of region-specific legal requirements and ethical guidelines. Researchers, regulatory affairs specialists, and clinical trial coordinators could be the targeted users. By simplifying the navigation of complex legal environments, the RDRN would facilitate smoother and more ethically compliant research processes, reducing delays and legal risks.
Data Sharing and Privacy Initiatives	With big data playing an increasingly critical role in research, funding could support the creation of protocols and best practices that ensure the ethical use and sharing of data, while respecting patient privacy and the specific confidentiality concerns associated with rare diseases in the newborn screening space.	Develop a “SecureDataShare Protocol” to ensure the ethical use and sharing of data in newborn screening research, with a strong emphasis on protecting patient privacy. Key features include encryption standards for data storage and transfer, protocols for anonymizing sensitive information, consent management systems that allow patients to control their data usage, and guidelines for ethical data sharing practices. Collaborators could be IT specialists, bioethicists, data protection officers, and patient advocacy groups. This initiative would enhance trust in research by ensuring that patient data are handled with the highest ethical standards, while still enabling valuable scientific collaborations.
Policy Development and Advocacy	Developing comprehensive policy frameworks that integrate ethical, legal, and social considerations into all aspects of newborn screening programs. Advocating for sustained funding and resources to support ELSI research and the implementation of best practices in newborn screening. Facilitating collaboration between policymakers, researchers, healthcare providers, and patient advocacy groups to ensure that policies are well-rounded and effectively implemented.	A task force dedicated to developing and advocating for policies that integrate ethical, legal, and social considerations into all aspects of newborn screening programs. Activities include conducting policy analysis, drafting policy recommendations, organizing advocacy campaigns, and facilitating stakeholder meetings. Members could be policymakers, researchers, healthcare providers, patient advocates, and legal experts. This task force would ensure that newborn screening programs are guided by comprehensive, well-rounded policies that reflect the needs and concerns of all stakeholders, leading to more effective and ethically sound screening practices.

## Data Availability

All data are available within this article.

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
