# Peer review of "Charting the Ethical Frontier in Newborn Screening Research: Insights from the NBSTRN ELSI Researcher Needs Survey"

_2409-515X, 2024, doi:10.3390/ijns10030064_

Round 1

Reviewer 1 Report

Comments and Suggestions for Authors

This manuscript describes the result of a survey of individuals involved in state newborn screening programs regarding ELSI issues.  The survey was conducted by the NBSTRN.  The issues are timely and the survey appears to have been well-conducted.

I have several suggestions to improve clarity.

1) The authors note that the NBSTRN was discontinued this year,  Yet, on lines 79-81, they state an intention of developing additional tools and resources.  How will such development be pursued without the NBSTRN?

2) The manuscript shifts back and forth between outlining the ELSI issues in NBS and ELSi issues relevant to NBS research and researchers.  Many of the ELSI issues outlined are relevant to NBS public health programs, particularly if they are adding a new condition.  In addition, pilot data for considerations regarding the RUSP often come from one or more state programs that have adopted a new condition, not from formal research protocols.  The point is that the authors should clarify what constitutes research (and researchers) in this context and thus who this paper is designed to inform.  It is also fine if the paper is targeting the broader community of stakeholders involved in NBS decision-making, not just investigators conducting pilot research.  The point is, more clarity at the beginning and consistency through the manuscript would be helpful.

3) The authors might wish to note some of the challenges to ELSI research in this domain.  As we know, NBS is conducted by state public health departments, yet NBS research, the NBSTRN, and decision-making about the RUSP are conducted by organizations at a national level.  Is ELSI support for NBS researchers sufficient if the primary decisions about implementation are being conducted by individual state programs?  To what extent are the results of this study, and the ELSI resources advocated, relevant to state program decision-makers and not just researchers?  To what extent should ELSI issues be addressed in collaboration with state programs?

4) The introduction seems a bit long and a bit redundant regarding the need for ELSI support for NBS.  The issues outlined on lines 138 - 160 might be included as a table.

5) The proposed funding initiatives outlined in lines 306-336 don't suggest what organization(s) would be funding these initiatives.  Some comment might be added to indicate what organizations might be best situated to fund these resources.

Reviewer 2 Report

Comments and Suggestions for Authors

This work is interesting and is focused on ethical concerns about NBS. However, the paper is relevant in the US context and maybe some overview of ethical situation in other countries should be important.

- In Materials and methods, the model of survey should be add in supplemental files is possible

- Results : may be, some importants findings should be presented in a figure in complement of the globals results in the table.

- Results : in the part "themes shared by participants", may be the number of contributors should be add to each item.

- In the discussion, authors could add a part on the global interest of the future tools that they proposed to the rest of the world.

Reviewer 3 Report

Comments and Suggestions for Authors

Summary

The paper discusses the crucial need for early Ethical, Legal, and Social Implications input in Newborn Screening research and pilot studies. It surveys a large group of NBS researchers and other stakeholders to identify ELSI needs and resources necessary for integrating ELSI considerations early and seamlessly into NBS research. It clearly outlines key ELSI topics that should be addressed early in current and future NBS research and pilots. The paper raises awareness of ELSI requirements among NBS researchers, aligning well with the readership of IJNS. A survey was conducted among researchers involved in NBS related research or pilot studies to gain insights into ELSI needs and resources, as well as the challenges participants face in their NBS research. Additionally, the paper describes the history of the NBSTRN ELSI Advantage Tool, highlights the need for ELSI resources among NBS researchers, and proposes new ELSI tools and resources based on survey findings.

Value of the paper

The paper's value lies in its focus on identifying ELSI needs among NBS researchers, with a clear goal (line 174-175) and target population of researchers and those involved in NBS research including expansion of NBS. Early ELSI awareness is advocated, supported by a survey that received significant response from the relevant research community.

Major comments

Introduction:

Line 174-175: The paper's goal is described as identifying NBS researchers’ ELSI needs through a survey targeting researchers involved in NBS-related studies.

Line 108: Paragraph 1.3 summarizes existing literature and ELSI needs in NBS research, suggesting to include this section earlier into the introduction to capture the target audience's attention.

Line 148: References to social and ethical issues are more than 13 years dated (2008-2011), recommending the use of more recent resources.

Line 51: The position of the NBSTRN ELSI tool is not completely clear throughout the paper. Although its history is elaborated in the introduction, it is not mentioned by the participants as a source of information in question 5 (line 226). The discussion concludes with the need for more funding for other tools, neglecting to discuss lessons learned from the existing valuable tool, such as dissemination, access and information presentation methods, briefly touched upon in line 304.

Materials and Methods:

Line 190: Some questions are included in table 1 and throughout the results section. Providing a comprehensive overview of the survey questionnaire, possibly in supplemental data, would be informative.

Line 204: Additional information on data flow would be informative. Were all respondents included? Were all surveys fully answered? More details on the selected of the questions would provide insight such as three demographic questions, x on information needs, and x were open-ended questions. The method section lacks an analysis description of the ranking of ELSI learning preferences and the selection process for open-ended question results (described later in line 230-271).

Results:

Line 225: Table 1 presents learning tool preferences (discussed in line 301-303). More information on the ranking determination process would aid interpretation. Insight into the relation of learning tool preferences to other factors, such as research affiliations or years involved in newborn screening research, would be valuable for designing future tools.

Line 226: Paragraph 3.1 lists nine resources survey respondents consult. Additional background information is necessary to interpret their value. Were these nine items grouped or the only answers provided?

Line 244: NBSTRN ELSI resources were not listed in question 5 as sources of information and guidance mentioned by participants. However, they were the subject of question 6. Clarification is needed regarding the resource's status as an information source for the participants. Information on the number of percentage of respondents who answered this question and found the resource helpful, 30% as mentioned in line 303, would enhance the results section.

Line 256: The open-ended question on ELSI challenges nicely describes one of the survey's purposes “to assess the ethical and legal challenges that NBS researchers face”. More information on the number of respondents reporting issues, the researchers affiliation/background and the nature (grouped, selection, all) of the reported items would aid interpretation.

Lines 273-286: Key findings are well-described.

Discussion:

Line 288: The first part of the discussion focuses on survey results before transitioning to funding initiatives. Bridging the survey findings with proposed funding initiatives would aid coherence. Additionally, discussing how to improve the usage and uptake of existing tools (line 301-304) beyond funding would be beneficial. For example chances for better dissemination of existing tools, exploring the usefulness of tools from other fields such as other population screening programmes. Addressing any limitations in the study according to the authors would be useful.

Line 351: Not all survey questions are included in the paper, and there is no ethics statement.

Round 2

Reviewer 3 Report

Comments and Suggestions for Authors

Summary

The revised version of the paper clearly presents insights gathered from a broad survey conducted among ELSI researchers related to newborn screening (NBS) research. The authors revised the paper according to the suggested improvements.

The aim of the study is to achieve a better understanding of ELSI issues in NBS research encountered by the broader NBS community. The revised version has a much better focus on the survey, and the results are more clearly presented than in the previous version. The survey results provide insight into the needs for ELSI resources among NBS researchers. The authors have distilled future efforts of importance for ELSI issues in NBS research from the survey results. The illustration of future efforts defined by the authors provides a broad overview of key ELSI areas, including examples. This latter part is better connected to the survey findings and the study's objectives in this revised version.

The revised version has been largely rewritten, including adjustments to the author list. The value of the paper remains unchanged, as it focuses on identifying ELSI needs among NBS researchers. Early ELSI awareness is advocated, supported by a survey that received significant responses from the relevant research community. Key takeaways are formulated by the authors to guide future efforts in ELSI and NBS research.

Minor Comments

 Figure 1: The resolution for the review process is insufficient to read everything clearly. However, the description in the text and the questions in the appendix provide enough confidence that this is an informative and clear figure. The reader can clearly link the results in the text to the appropriate figures.

Table 2: A header for the table is missing. The text does clearly refer to the focus in line 208: "Table 2 lists suggested future efforts and provides examples to illustrate how targeted efforts can address ELSI considerations in NBS research." Please add a comparable header.

Line 209 and Line 221 (REF): Reference missing.

Lines 203-225, Explanation of Future Efforts: The survey study is clearly described, and the conclusions are clear up to line 203. From line 203 onward, the authors transition to future efforts, which are extensively described in Table 2. This is a good summary, but it could be introduced somewhat more with a few sentences. For example a brief explanation from the authors about the basis of these future efforts—whether from the survey, years of experience in NBSTRN network, documentation from this network, or papers related to NBSTRN—would help to better highlight the value of these key aspects. This can help those not directly involved in NBSTRN to understand where these nine key areas come from, likely from the survey information and the authors' involvement in NBSTRN efforts as described in the introduction lines 102-117. A brief explanation in lines 203-209 would be valuable. I trust the authors can clarify this without the need for further review.

Author Response

Comment 1: Table 2: A header for the table is missing. The text does clearly refer to the focus in line 208: "Table 2 lists suggested future efforts and provides examples to illustrate how targeted efforts can address ELSI considerations in NBS research." Please add a comparable header.

Response 1: Thank you for the helpful suggestion. The following header was added: Table 2. Nine Suggested Areas for Future Efforts to Advance ELSI in NBS Research for the Worldwide NBS Community

Comment 2: Line 209 and Line 221 (REF): Reference missing.

Response 2: Thank you. Added the following reference: Bick D, Ahmed A, Deen D, Ferlini A, Garnier N, Kasperaviciute D, Leblond M, Pichini A, Rendon A, Satija A, Tuff-Lacey A, Scott RH. Newborn Screening by Genomic Sequencing: Opportunities and Challenges. Int J Neonatal Screen. 2022 Jul 15;8(3):40. doi: 10.3390/ijns8030040. PMID: 35892470; PMCID: PMC9326745.

Comment 3: Lines 203-225, Explanation of Future Efforts: The survey study is clearly described, and the conclusions are clear up to line 203. From line 203 onward, the authors transition to future efforts, which are extensively described in Table 2. This is a good summary, but it could be introduced somewhat more with a few sentences. For example a brief explanation from the authors about the basis of these future efforts—whether from the survey, years of experience in NBSTRN network, documentation from this network, or papers related to NBSTRN—would help to better highlight the value of these key aspects. This can help those not directly involved in NBSTRN to understand where these nine key areas come from, likely from the survey information and the authors' involvement in NBSTRN efforts as described in the introduction lines 102-117. A brief explanation in lines 203-209 would be valuable. I trust the authors can clarify this without the need for further review.

Response 3: Thank you for the helpful suggestion. We revised starting at line 204: These future efforts to support ELSI in NBS research will be important as the number of diseases, treatments, and technologies increase and the complexities surrounding ELSI issues continue to grow. In addition, groups around the world researching ways to integrate genomics to screen, diagnose, and design interventions in newborns and children and this heightens the need for robust, coordinated, and widely disseminated ELSI efforts and policies [46]. Informed by the fifteen years of discussions, collaborations, and workgroups of the NBSTRN, Table 2 lists nine suggested key areas for future efforts and provides examples to illustrate how targeted projects can address ELSI considerations in NBS research by leveraging patient-centric resources, participatory research models, public education and engagement, and collaborative education programs. The NBSTRN network of diverse stakeholders across the NBS community from public health partners to clinicians, patients, parents, families, researchers, and advocacy groups continually highlighted the importance of a focus on privacy, informed consent, interdisciplinary collaboration, social acceptance, and ongoing education. This broad approach to ELSI in NBS research coupled with innovative initiatives that considers local, regional, and county-wide policies and practices will help to ensure that ELSI aspects are comprehensively integrated into NBS research around the world. In the US, continued collaboration between the National Institutes of Health (NIH), the Centers for Disease Control and Prevention (CDC), and the Health Resources and Services Administration (HRSA) will be vital in supporting these efforts to advance ELSI in NBS research and ensure the successful implementation of evolving best practices. Each of these agencies brings unique strengths and expertise to the table, creating a comprehensive approach that addresses scientific discovery, translational research, public health implementation and surveillance, and clinical and community-based care across the lifespan delivery aspects (47). In conclusion, the NBSTRN ELSI survey results demonstrate that the NBS community recognizes the important of ELSI in NBS research infrastructure and calls upon all stakeholders and federal partners to invest in a future where rare disease research is prioritized and supported across research, clinical care, and state NBS programs.